# Personal and Work-Related Burnout Is Associated with Elevated Diastolic Blood Pressure and Diastolic Hypertension among Working Adults in Chile

**DOI:** 10.3390/ijerph20031899

**Published:** 2023-01-19

**Authors:** Yinxian Chen, Diana Juvinao-Quintero, Juan Carlos Velez, Sebastian Muñoz, Jessica Castillo, Bizu Gelaye

**Affiliations:** 1Department of Epidemiology, Harvard T.H. Chan School of Public Health, Boston, MA 02115, USA; 2Departamento de Rehabilitación, Hospital del Trabajador, Asociación Chilena de Seguridad, Santiago 8320000, Chile; 3Department of Psychiatry, Harvard Medical School, Boston, MA 02115, USA; 4The Chester M. Pierce, MD Division of Global Psychiatry, Massachusetts General Hospital, Boston, MA 02114, USA

**Keywords:** personal burnout, work-related burnout, blood pressure, hypertension, working adults

## Abstract

We aimed at investigating the association of personal and work-related burnout with blood pressure and hypertension among working adults in Chile. We conducted a cross-sectional study among 1872 working adults attending the Hospital del Trabajador in Santiago, Chile, between September 2015 and February 2018. The Copenhagen Burnout Inventory was used to assess personal and work-related burnout. Blood pressure was measured by medical practitioners. Multivariable linear and logistic regressions were used to estimate the association of burnout status with systolic blood pressure (SBP), diastolic blood pressure (DBP), and hypertension. After adjusting for confounders, participants with both types of burnout had a 1.66 (95% confidence interval [CI]: 0.02–3.30) mmHg higher mean DBP than those without burnout. The odds of isolated diastolic hypertension among the participants with only personal burnout and both types of burnout were 2.00-fold (odds ratio [OR] = 2.00; 95% CI: 1.21–3.31) and 2.08-fold (OR = 2.08; 95% CI: 1.15–3.78) higher than those without burnout. The odds of combined systolic/diastolic hypertension among the participants with only work-related burnout increased by 59% (OR = 1.59; 95% CI: 1.01–2.50) compared with those without burnout. Both work-related and personal burnouts were associated with increased DBP and odds of diastolic hypertension among working adults in Chile.

## 1. Introduction

Burnout is a state resulting from individuals being exposed to prolonged psychosocial stress [1] and presenting core features such as physical and psychological fatigue and exhaustion [2]. Although previous studies have mainly been applied in describing the status of working adults under exposure to working stress, burnout can be due to other attributes in the personal life of working adults, such as health problems and family demands.

Hypertension is a leading modifiable cause of premature mortality and a significant contributor to the global burden of cardiovascular disease (CVD) [3], contributing to 26.4% of all causes of death [4]. According to the Chilean National Health Survey, 30.8% of Chileans had hypertension in 2017, with men (31.2%) having a slightly higher prevalence than women (30.3%) [5]. For young and middle-aged adults (<60 years), isolated diastolic hypertension (IDH: SBP < 140 mmHg and DBP ≥ 90 mmHg) is more prevalent than isolated systolic hypertension (ISH: SBP ≥ 140 mmHg and DBP < 90 mmHg) [6]. Among them, elevated DBP mainly contributes to the risk of CVD [7]. In contrast, older adults (≥60 years) are more likely to have combined systolic/diastolic hypertension (SDH: SBP ≥ 140 mmHg and DBP ≥ 90 mmHg) and isolated systolic hypertension (ISH: SBP ≥ 140 mmHg and DBP < 90 mmHg) [6]. In this case, increased SBP replaces DBP as the predominant predictor of CVD [7]. Given that hypertension is highly prevalent in Chile, it is warranted to explore populations susceptible to hypertension to decrease the burden of CVD in Chile.

Burnout has shown strong potential as a target for hypertension prevention. Previous studies have indicated that burnout is associated with multiple health consequences, ranging from health behaviors (smoking [8], alcohol consumption [8], and substance abuse [8]), psychiatric disorders [9,10], and insomnia [9] to physical disorders (type 2 diabetes [9] and hypercholesterolemia [9]) that are well-established risk factors for hypertension [11,12]. However, the current evidence of burnout with hypertension among working adults is inconclusive. More studies have shown that burnout is associated [13,14,15,16,17,18] with higher odds or a higher proportion of hypertension than those showing no association [19,20,21]. The association between burnout and blood pressure is also heterogeneous, going from no association with SBP and DBP [22] to being associated with SBP/DBP [23] or with mean arterial pressure (MAP) [24]. The current evidence is also narrow in terms of the attributes of burnout, with only one study assessing generic burnout [17], while others have only focused on work-related burnout. Nearly all the evidence has been derived from the human service sectors, with only one exception focusing on mine workers [20]. The quality of the current evidence is also problematic, as some studies did not take into account the presence of confounding variables under their observational study design [13,14,16,20,25,26].

Given the gap described above, a study that systematically assesses the association between burnout and hypertension among working adults is necessary to provide new evidence and promote additional health strategies for addressing the cardiometabolic consequences of high burnout among working adults. Thus, we conducted a cross-sectional study in an ongoing cohort, recruiting working adults in Santiago, Chile, to examine the association of burnout with blood pressure and the occurrence of hypertension.

## 2. Materials and Methods

### 2.1. Study Population

The present cross-sectional study drew participants from the Stress, Pain, Sleep, and Neuropsychiatric Disorders (SPLENDID) study conducted in a workers’ compensation hospital system in Chile. Details about the SPLENDID study, including the study procedures, have been previously described [27]. Briefly, the SPLENDID study aimed to examine the prevalence and correlations of pain, work-related stress, and neuropsychiatric outcomes among working adults in Chile with the goal of developing workplace wellness programs. Between September 2015 and February 2018, we collected data among patients attending the Hospital del Trabajador in Santiago, Chile. The Hospital del Trabajador is the largest workers’ compensation hospital and the referral center for trauma and professional diseases of the Asociación Chilena de Seguridad, with approximately 2.5 million affiliated workers. Individuals were eligible if they were working adults attending the hospital for the following types of injuries: burns, bone fractures, spinal cord and mild brain injury, and soft tissue injuries of various etiologies. All participants provided written informed consent prior to participation. All procedures in this study were approved by the institutional review boards of the Hospital del Trabajador, Santiago, Chile, and the Office of Human Research Administration, Harvard T.H. Chan School of Public Health, Boston, MA, USA.

### 2.2. Exposure

While the Maslach Burnout Inventory (MBI) [2] is the most widely used tool for assessing burnout, the personal attributes of burnout outside the work setting are not captured in it. Therefore, we chose the Copenhagen Burnout Inventory (CBI), a 19-item scale measuring burnout in 3 domains: personal burnout (6 items), work-related burnout (7 items), and client-related burnout (6 items) [28]. The CBI has been shown to have good reliability and validity among the Spanish-speaking population [29,30]. Since the present study was conducted among the general working population, the client domain was not included due to its inapplicability in non-human-service sectors.

The personal burnout scale included six items. The responses for each item were given on a five-point Likert scale, ranging from 0 to 4, which corresponded to never/almost never, seldom, sometimes, often, and always. The responses were then transformed into scores of 0, 25, 50, 75, and 100, consistent with the original questionnaire. The score for personal burnout was the average score of 6 items and ranged from 0 to 100. We defined scores of ≥50 as having personal burnout [31,32]. The work-related burnout subscale contained seven items. The responses for each item were given on a five-point Likert scale: very-high degree, high degree, somewhat, low degree, and very-low degree. Similar to personal burnout, the responses were transformed into scores of 0, 25, 50, 75, and 100. The total score for work-related burnout was the average score of the 7 items and ranged from 0 to 100. We defined scores of ≥50 as having work-related burnout [31,32]. To examine the independent and joint effects of the two burnout domains, we categorized participants into four groups: (1) no burnout (both subscales < 50); (2) only personal burnout (only personal subscale ≥ 50); (3) only work-related burnout (only work-related subscale ≥ 50); and (4) both burnouts (both subscales ≥ 50).

### 2.3. Outcome

After participants completed the structured interviews, their blood pressure was measured. Two additional blood pressure measurements were taken with 3 min elapsing between successive measurements. Following the World Health Organization’s recommendation, the mean systolic blood pressure (SBP) and diastolic blood pressure (DBP) from the second and third measurements were analyzed. Hypertension was defined as mean SBP ≥ 140 mmHg or mean DBP ≥ 90 mmHg based on the Seventh Joint National Committee on prevention, detection, evaluation, and treatment of high blood pressure (the JNC 7) [33]. Subtypes of hypertension [34], including isolated diastolic hypertension (IDH: SBP < 140 mmHg and DBP ≥ 90 mmHg), isolated systolic hypertension (ISH: SBP ≥ 140 mmHg and DBP < 90 mmHg), and combined systolic/diastolic hypertension (SDH: SBP ≥ 140 mmHg and DBP ≥ 90 mmHg), were used as the outcomes for the secondary analysis.

### 2.4. Covariates

We used structured questionnaires to collect participants’ sociodemographic and behavioral characteristics. Sociodemographic characteristics included sex (men vs. women), age when interviewing (continuous and <60 years vs. ≥60 years), country (Chile vs. other), and highest education level (elementary school, high school, and college or technical training). Behavioral characteristics included body mass index (BMI) (<18.5 kg/m^2^, 18.5–24.9 kg/m^2^, 25–29.9 kg/m^2^, and ≥30 kg/m^2^), smoking status (non-smoker, past smoker, and current smoker), alcohol consumption (never, five days/week, and ≥5 days/week), and physical activity, which was assessed by a Global Physical Activity Questionnaire (GPAQ), where participants having a Metabolic Equivalent (MET) ≥ 600 min per week were considered physically active [35] (inactive vs. active). We also measured participants’ occupational characteristics, including their work sector (construction, commercial, finance, public services, manufacturing, transportation, and others), work type (administrative, professional, manual worker, salesperson, teacher, technician, and others), and work shift (7 a.m.–7 p.m., 2 p.m.–12 a.m., and 11 p.m.–8 a.m.). All covariates were considered potential confounders in this study based on the previous evidence showing that they were associated with objective exposure (burnout) and outcome (blood pressure and hypertension).

### 2.5. Statistical Analysis

We interviewed a total of 2000 participants. Of these, based on the complete case analysis (CCA), sixty-six participants were excluded due to missing data on blood pressure, seven were excluded due to missing data on the CBI, and seventy-five were excluded due to missing data on relevant covariates. We included a total of 1872 participants in the final study.

We first explored the frequency distributions of participants’ sociodemographic, behavioral, and occupational characteristics. We summarized the data using numbers and percentages (%) for the categorical variables and reported group-specific means and standard deviations (SD) for the continuous variables. We used the Chi-square test or Fisher’s exact test for the categorical variables and ANOVA (normally distributed and homogeneous variance) or the Kruskal–Wallis test (non-normally distributed or heterogeneous variance) for the continuous variables to determine bivariate differences. We considered confounding variables a priori based on their hypothesized relationship with burnout and elevated BP. The covariates of interest were sex, age, country, highest education level, BMI, physical activity, work sector, work type, and work shift. For the primary analysis, we used multivariable least squares linear regression procedures to assess the association of burnout status with mean SBP and DBP before and after accounting for prior-selected putative confounders, reporting crude and adjusted coefficients (βs), 95% confidence intervals (CIs), and the *p*-value (P). We completed analyses for each outcome separately. In addition, we used logistic regression procedures to estimate the odds ratios (ORs) and 95% Cis of hypertension in relation to burnout status. Given the qualitative differences in the endorsement of personal burnout by sex [32,36], we repeated the same procedures in the primary analyses, stratifying by sex, and estimating the sex-specific βs, ORs, and 95% CIs. We also used polytomous logistic regression to estimate the ORs and 95% CIs of the association of burnout status with subtypes of hypertension (IDH vs. normal, ISH vs. normal, and SDH vs. normal). Two-side *p* < 0.05 was considered statistically significant. All analyses were performed using R version 4.1.2.

## 3. Results

Table 1 shows the characteristics of the study participants (*n* = 1872). The mean age (SD) of the participants was 45.8 (13.7), and most of them were men (73.7%), born in Chile (93.9%), and had at least a high school education (56.0%). About two-thirds of participants reported lifetime smoking (30.3% for past smokers and 30.6% for current smokers), while 5.5% reported alcohol consumption nearly every day.

The prevalence of hypertension was 27.2% (7.2% for IDH, 9.2% for ISH, and 10.8% for SDH). Table 1 also shows the participants’ characteristics by burnout. Those without burnout were more likely to be younger men working in construction, commercial, or professional sectors. They were more likely to report alcohol consumption below five days per week and had higher education levels compared with participants with burnout (*p* < 0.005). Those without burnout were also more likely to have a lower prevalence of hypertension and SDH (*p* < 0.02).

Table 2 shows that both personal and work-related burnouts were associated with a higher mean DBP. After adjusting for confounders, the participants with both types of burnout had a 1.66 (95% CI: 0.02–3.30) mmHg higher mean DBP than those without burnout. The participants with only personal (β = 0.53; 95% CI: −0.83–1.90) or work-related burnout (β = −0.10; 95% CI: −1.58–1.38) did not show an elevated mean DBP compared with the reference group. All types of burnout were not associated with an increased mean SBP. The odds of hypertension among the participants with only work-related burnout increased by 40% (OR = 1.40; 95% CI: 0.99–1.96) compared with those without burnout, with marginal significance, while personal burnout (OR = 1.21; 95% CI: 0.88–1.67) and both burnouts (OR = 1.31; 95% CI: 0.89–1.90) were not statistically significantly associated with elevated odds of hypertension.

Appendix A shows that having both burnouts was associated with a 2.11 (95% CI: 0.06–4.15) mmHg higher DBP compared with no burnout among men after adjusting for putative confounders. Men with both burnouts and only work-related burnout had 1.65-fold (95% CI: 1.04–2.58) and 1.57-fold (95% CI: 1.07–2.28) higher odds of hypertension compared with those without burnout. However, all types of burnout were not statistically significantly associated with SBP, DBP, and the odds of hypertension among women (Appendix A).

The association between burnout status and subtypes of hypertension is presented in Table 3. After accounting for putative confounders, the odds of IDH among the participants with only personal burnout and both burnouts were 2.00-fold (OR = 2.00; 95% CI: 1.21–3.31) and 2.08-fold (OR = 2.08; 95% CI: 1.15–3.78) higher than those without burnout. The odds of SDH among the participants with only work-related burnout increased by 59% (OR = 1.59; 95% CI: 1.01–2.50) compared with those without burnout. All types of burnout were not associated with increased odds of ISH.

## 4. Discussion

In the present study of Chilean working adults, 27.2% of the participants had hypertension, with 7.2% having IDH, 9.2% having ISH, and 10.8% having SDH. We found that the participants with both personal burnout and work-related burnout were associated with elevated DBP after adjusting for prior putative confounders. The results were consistent among men but not women. We also found that odds of IDH increased among the participants with only personal burnout and both burnouts compared with those without burnout. The elevated odds of SDH were found among the participants with only work-related burnout. Little evidence was found of an association of any status of burnout with elevated SBP and ISH.

Our study extended the findings from prior studies [13,14,15,16,17,18,19,20,21,22,23,24], documenting that exposure to personal and work-related burnout was related to an increased DBP and higher odds of IDH. These findings are vital for working adults. Although previous evidence has indicated that an increase in SBP or having ISH or SDH is the dominant predictor of cardiovascular outcomes, and that changes in DBP and the occurrence of IDH is considered less important in preventing cardiovascular risk [37], this circumstance is age-dependent [7]. The results from The Framingham Heart Study showed that elevated DBP was the dominant predictor of coronary heart disease (CHD) among individuals <50 years, and the risk of CHD by elevated DBP and SBP was comparable among those <60 years [7]. Another piece of evidence from UK Biobank indicated that for individuals <60 years, the hazard of composite CVD events (nonfatal MI, nonfatal ischemic stroke, nonfatal hemorrhagic stroke, and CVD death) increased among those with IDH compared with those without IDH [38]. One longitudinal study recruiting adults from all age ranges (≥18 years, median = 53 years) showed that increased DBP and IDH were still independently associated with a higher hazard of adverse CVD events (first episode of myocardial infarction, ischemic stroke, or hemorrhagic stroke) [39], although yielding a smaller effect size than elevated SBP and ISH. Given that most of our participants were working adults (mean age = 45.7 years, with 82.6% <60 years), the burden of increased DBP and the odds of IDH might contribute to the younger occurrence of CVD events, yielding an inability to work and a poor quality of life. Moreover, IDH was associated with future ISH and SDH [40], leading to higher risk and worse prognosis of CVD events in later life [7]. Therefore, interventions to alleviate burnout from not only the working environment but also personal life may help decrease the risk of CVD among working adults by improving their DBP.

The substantial variability in our results and the existing evidence might account for the differences in the measurement tools of burnout and the population recruited in the studies [22]. Previous studies have mainly focused on burnout derived from work settings [13,14,15,16,17,19,20,21,22,23,24] and individuals in the human service sector (health professionals [13,14,15,19], university professors [16], company managers [21], and workers in elderly welfare facilities [17]). Only one study focused on personal burnout [17], and one was conducted in other sectors (mine workers [20]). We extended the current evidence to a new magnitude and broader sources of burnout and the population by assessing personal burnout and recruiting working adults from multiple sectors and occupational types.

The potential mechanism that burnout may lead to increased blood pressure and risk of hypertension usually accounts for the autonomic nervous system (ANS) and the hypothalamic–pituitary–adrenal (HPA) axis [41], which are the primary systems responding to stressors in humans. Burnout has been characterized as a prolonged psychosocial stress exposure, yielding a persistent unbalance between demands and resources [1]. This mismatch may lead to the sustained activation of the sympathetic nervous system (SNS) and suppression of the parasympathetic system (PNS) [41]. The level of catecholamines (such as adrenaline and noradrenaline) would thus persistently increase, raising the DBP and subsequently increasing the risk of IDH by causing vasoconstriction and constant peripheral resistance. The high DBP related to the prolonged activation of the SNS may also be due to an inactive reaction of the HPA axis. Existing evidence has found that hypocortisolism occurs in individuals with burnout [42], which could decrease the negative feedback from cortisol to corticotropin-releasing hormone (CRH), thus raising CRH levels persistently. CRH is a hormone that can activate the SNS [43], leading to prolonged high DBP. Sustained high peripheral resistance can increase atherosclerosis risk, impairing the structure of large arteries and making them less elastic. Together with the effect of age on increasing the stiffness of large arteries [44], IDH related to burnout may switch to SDH or ISH due to the elevation of SBP when individuals grow older [40]. Our results might manifest this progress to some extent. The participants with only personal burnout (with 17.9% ≥60 years) and both burnouts (with 14.0% ≥60 years) were younger and were associated with higher odds of IDH. In comparison, the participants with only work-related burnout (with 23.3% ≥60 years) were older and were related to higher odds of SDH. Burnout is also associated with hypercholesterolemia [9], type 2 diabetes [9], and a high risk of unhealthy behaviors such as smoking [8], alcohol consumption [8], and substance abuse [8], all of which are considered risk factors for increased blood pressure and hypertension [11,45].

Previous studies have shown that women have sustained excessive demands outside of their work [36] and are more likely to experience burnout compared with men [32], which was consistent with the current study, showing that women had a higher proportion of personal burnout (36.7% in women vs. 18.7% in men). However, all types of burnout, including personal burnout, did not have a statistically significant association with elevated blood pressure and odds of hypertension, which was different from the results in men. We do not have a clear explanation for this. The lack of a statistically significant association among women might be due to a lower underlying risk of hypertension than men; even though a larger proportion of women were exposed to personal burnout, they may still be less likely to have hypertension.

There were serval strengths in our study. First, we assessed burnout from both the working environment and personal life, accounting for the potential joint effect of burnout from different attributes. Second, to the best of our knowledge, this was the first study examining the association between burnout and hypertension subtypes, presenting a clearer image of how burnout might affect the occurrence of hypertension. Third, our study population comprised working adults from multiple types of occupations in different work sectors, increasing the generalizability of our results compared with previous studies.

### Limitations

As a cross-sectional study, the temporality of burnout and hypertension could not be determined. Additionally, we did not collect information on the participants’ prior diagnosis of hypertension and anti-hypertensive treatment. Whether the participants had hypertension was defined only by their blood pressure, which might lead to the misclassification of the outcome because we did not recognize hypertensive patients with normal blood pressure maintained by anti-hypertensive treatment. Moreover, prior diagnosis of hypertension and medical use may be associated with current blood pressure and hypertension, as well as the emotional burden of working adults, leading to higher levels of burnout. The lack of adjustment for those factors may yield residual confounding bias. For applying self-reported measurement tools on burnout and most covariates, measurement errors might have caused bias in the results. As we drew the participants from a worker’s compensation hospital, the results of our study might not be generalizable to working adults from other bases. Finally, although the regression models were adjusted for possible confounding factors, residual confounding could have possibly remained due to these unmeasured factors.

## 5. Conclusions

Both work-related and personal burnouts were associated with an increased DBP and odds of diastolic hypertension among working adults in Chile. Mental health support for burnout may effectively prevent hypertension among working adults. Future researchers should consider conducting prospective cohort studies or randomized trials to determine the temporality between burnout and hypertension.

## Figures and Tables

**Table 1 ijerph-20-01899-t001:** Characteristics of working adults in Santiago, Chile, by burnout status (*n* = 1872).

	All Participants	No Burnout	Only Personal	Only Work-Related	Both	
	(*n* = 1872)	(*n* = 1218)	(*n* = 268)	(*n* = 215)	(*n* = 171)	*p* ^a^
Sex, %						
Men	73.7	78.4	57.1	77.2	61.4	<0.0001
Age (years), %						
<60	82.6	83.3	82.1	76.7	86.0	0.08
≥60	17.4	16.7	17.9	23.3	14.0	
Age (years), mean (SD)	45.7 (13.7)	44.5 (14.2)	47.7 (12.7)	48.6 (13.1)	47.9 (11.2)	<0.0001
Country, %						
Chile	93.9	93.0	94.8	97.2	94.2	0.11
Others	6.1	7.0	5.2	2.8	5.8	
Highest education levels, %						
Elementary school	16.1	12.8	21.3	24.2	21.6	<0.0001
High school	56.0	57.6	52.2	52.1	55.6	
College or technical training	27.9	29.6	26.5	23.7	22.8	
Work sector, % ^b^						
Construction	11.9	13.7	9.3	7.9	7.6	0.0005
Finance	1.8	2.1	1.9	0.0	2.2	
Commercial	18.3	21.3	13.8	7.9	17.0	
Manufacturing	27.7	25.6	29.1	43.3	21.0	
Public services	21.9	19.6	27.2	23.7	27.5	
Transportation	8.7	8.8	7.1	10.2	8.8	
Others	9.7	8.9	11.6	7.0	15.8	
Work type, % ^c^						
Administrative	13.4	14.4	13.4	8.4	12.9	0.0005
Manual worker	60.0	59.0	62.3	66.0	55.6	
Professional	6.6	8.1	4.9	4.2	1.8	
Salesperson	2.4	1.8	3.0	2.3	5.8	
Technician	7.4	7.5	7.5	7.0	7.0	
Teacher	0.9	0.9	0.4	0.5	1.8	
Others	9.3	8.3	8.6	11.6	15.2	
Work shift, %						
7 a.m.–7 p.m.	94.1	94.9	91.0	94.9	92.4	0.12
2 p.m.–12 a.m.	3.9	3.4	6.7	2.8	4.1	
11 p.m.–8 a.m.	2.0	1.6	2.2	2.3	3.5	
Smoking status, %						
Non-smoker	39.1	39.2	39.2	37.2	40.4	0.12
Past smoker	30.3	32.0	26.9	31.6	22.2	
Current smoker	30.6	28.7	33.9	31.2	37.4	
Alcohol consumption, %						
Never	38.1	35.3	44.8	39.5	46.2	0.003
<5 days/week	56.6	59.2	50.0	58.1	46.8	
≥5 days/week	5.2	5.5	5.2	2.3	7.0	
Physical activity, %						
Inactive	63.6	60.8	70.5	76.7	56.1	<0.0001
Active	36.4	39.2	29.5	23.3	43.9	
BMI (kg/m^2^), %						
<18.5	0.6	0.7	0.7	0.5	0.6	0.26
18.5–24.9	24.5	25.8	22.4	24.7	18.7	
25–29.9	42.7	42.5	40.7	47.0	41.5	
≥30	32.2	31.0	36.2	27.9	39.2	
Mean SBP (mmHg), mean (SD)	126.5 (15.3)	126.3 (15.0)	125.8 (15.9)	128.1 (15.9)	126.9 (15.9)	0.33
Mean DBP (mmHg), mean (SD)	79.9 (10.6)	79.6 (10.4)	80.0 (10.9)	80.2 (10.6)	81.3 (11.2)	0.22
Hypertension, % ^d^						
Yes	27.2	24.7	29.5	35.8	31.0	0.003
IDH, % ^e^						
Yes	7.2	6.2	9.7	7.4	9.9	0.11
ISH, % ^f^						
Yes	9.2	8.9	8.6	11.6	8.8	0.62
SDH, % ^g^						
Yes	10.8	9.5	11.2	16.7	12.3	0.02

Abbreviations: BMI, body mass index; DBP, diastolic blood pressure; IDH, isolated diastolic hypertension; ISH, isolated systolic hypertension; SBP, systolic blood pressure; SD, standard deviation; SDH, combined systolic/diastolic hypertension. ^a^
*p* value was calculated by chi-square test or Fisher’s exact test for categorical variables, and ANOVA or Kruskal–Wallis test for continuous variables. ^b^ Included agriculture, education, security, cleaning services, administration, food service, automotive, mining, retired, gardener, electrical engineer, and maintenance. ^c^ Included chauffeur, conductors, concierge, security guard, food service, landlord, food distribution, telecommunication, machine operator, and cleaning services. ^d^ Hypertension was defined as mean SBP ≥ 140 mmHg or mean DBP ≥ 90 mmHg. ^e^ Isolated diastolic hypertension was defined as mean SBP < 140 mmHg and mean DBP ≥ 90 mmHg. ^f^ Isolated systolic hypertension was defined as mean SBP ≥ 140 mmHg and mean DBP < 90 mmHg. ^g^ Systolic/diastolic hypertension was defined as mean SBP ≥ 140 mmHg and mean DBP ≥ 90 mmHg.

**Table 2 ijerph-20-01899-t002:** Association between burnout status and mean systolic blood pressure, diastolic blood pressure, and hypertension among working adults in Santiago, Chile (*n* = 1872).

	Mean SBP		Mean DBP		Hypertension	
	β (95% CI)	*p*	β (95% CI)	*p*	OR (95% CI)	*p*
Crude model						
No burnout	Ref.	-	Ref.	-	Ref.	-
Only personal burnout	−0.42 (−2.45, 1.60)	0.68	0.43 (−0.96, 1.84)	0.54	1.27 (0.95, 1.70)	0.11
Only work-related burnout	1.85 (−0.37, 4.07)	0.10	0.64 (−0.89, 2.17)	0.41	1.70 (1.25, 2.31)	0.0007
Both	0.67 (−1.78, 3.12)	0.59	1.75 (0.06, 3.44)	0.04	1.37 (0.96, 1.93)	0.08
Fully adjusted model ^a^						
No burnout	Ref.	-	Ref.	-	Ref.	-
Only personal burnout	−0.80 (−2.68, 1.07)	0.40	0.53 (−0.83, 1.90)	0.44	1.21 (0.88, 1.67)	0.24
Only work-related burnout	0.00 (−2.05, 2.05)	1.00	−0.10 (−1.58, 1.38)	0.90	1.40 (0.99, 1.96)	0.05
Both	−0.25 (−2.52, 2.02)	0.83	1.66 (0.02, 3.30)	0.05	1.31 (0.89, 1.90)	0.17

Abbreviations: β, coefficient; CI, confidence interval; DBP, diastolic blood pressure; OR, odds ratio; Ref., reference group; SBP, systolic blood pressure. ^a^ Adjusted for sociodemographic (sex, continuous age, country, and highest education levels), behavioral (BMI, smoking status, alcohol consumption, and physical activity), and occupational characteristics (work sector, work type, and work shift).

**Table 3 ijerph-20-01899-t003:** Association between burnout status and subtypes of hypertension among working adults in Santiago, Chile (*n* = 1872).

	Normal vs. IDH		Normal vs. ISH		Normal vs. SDH	
	OR (95% CI)	*p*	OR (95% CI)	*p*	OR (95% CI)	*p*
Crude model						
No burnout	Ref.	-	Ref.	-	Ref.	-
Only personal burnout	1.66 (1.04, 2.66)	0.04	1.02 (0.64, 1.65)	0.92	1.25 (0.82, 1.93)	0.30
Only work-related burnout	1.40 (0.79, 2.47)	0.25	1.52 (0.95, 2.44)	0.08	2.06 (1.36, 3.12)	0.0006
Both	1.74 (0.99, 3.04)	0.05	1.07 (0.60, 1.90)	0.82	1.41 (0.85, 2.33)	0.18
Fully adjusted model ^a^						
No burnout	Ref.	-	Ref.	-	Ref.	-
Only personal burnout	2.00 (1.21, 3.31)	0.007	0.83 (0.50, 1.38)	0.48	1.19 (0.75, 1.88)	0.47
Only work-related burnout	1.50 (0.82, 2.72)	0.19	1.13 (0.68, 1.88)	0.64	1.59 (1.01, 2.50)	0.04
Both	2.08 (1.15, 3.78)	0.02	0.91 (0.50, 1.67)	0.78	1.30 (0.76, 2.23)	0.34

Abbreviations: CI, confidence interval; IDH, isolated diastolic hypertension; ISH, isolated systolic hypertension; OR, odds ratio; Ref., referent group; SDH, combined systolic/diastolic hypertension. ^a^ Adjusted for sociodemographic (sex, continuous age, country, and highest education levels), behavioral (BMI, smoking status, alcohol consumption, and physical activity), and occupational characteristics (work sector, work type, and work shift).

## Data Availability

The data that support the findings of this study are available from the corresponding author upon reasonable request.

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
