# Peer review of "Personal and Work-Related Burnout Is Associated with Elevated Diastolic Blood Pressure and Diastolic Hypertension among Working Adults in Chile"

_ijerph, 2023, doi:10.3390/ijerph20031899_

Round 1

Reviewer 1 Report

The aim of this article "is to investigate the association between personal and work-related burnout with blood pressure levels and hypertension among Chilean workers". I really enjoyed reading the article, it is well written and clear in its purposes. I assess that it has publication merit, although two important aspects limit the scope of relevance and discussion of the findings and reduce the strength of the study: the cross-sectional design and the lack of adjustment for the use of medications to control or that interfere with blood pressure, nor the report of a previous medical diagnosis of hypertension.

Below I describe questions and suggestions.

One of the aspects that most caught my attention is the difference by gender in the frequency of personal burnout. Unlike the other classifications (Table 1), personal burnout seems to be more frequent among women than among men. This aspect makes me recommend that analyzes be carried out stratifying by sex and/or evaluating the modifying role of gender in the association. Several studies have shown that burnout is more frequent among women (Khasne RW, Dhakulkar BS, Mahajan HC, Kulkarni AP. Burnout among Healthcare Workers during COVID-19 Pandemic in India: Results of a Questionnaire-based Survey. Indian J Crit Care Med. 2020 Aug;24(8):664-671). In addition, several studies point to a female overload in demands outside work, especially with personal life and family care (Griep RH, Silva-Costa A, Chor D, Cardoso LO, Toivanen S, Fonseca MJMD, Rotenberg L. Gender, work -family conflict, and weight gain: four-year follow-up of the Brazilian Longitudinal Study of Adult Health (ELSA-Brasil).Cad Saude Publica.2022 May 9;38(4):EN066321.

Was there sample size planning? Among those eligible, what was the participation rate in the study? Was there self-selection bias?

Do the authors have evidence of psychometric burnout scales in the context of the study? Was there a psychometric adaptation of the instrument in the context of the study?

Some adjustment variables could be considered mediators between exposure and outcomes (mainly behavioral variables). Some variables were not shown to be associated with exposure in Table 1 and even so were included in the adjustment model. How was the contribution of each adjustment variable in the model evaluated? Could you present the operational theoretical model that guided the choice of adjustment variables?

In the discussion, I suggest deepening the different aspects that relate between personal burnout and blood pressure levels and hypertension and work-related and blood pressure levels and hypertension. What aspects could explain a greater importance of personnel or work depending on the type of outcome evaluated. What are the hypotheses of the authors to explain these findings. The form presented only compares with other studies, without further discussion of such interesting findings.

Reviewer 2 Report

I have read your manuscript with great interest and attention, which in my opinion is extremely interesting. The topic of burnout is really topical and represents a problem not only at work but also and above all of public safety and health. I have to ask you for some revisions.

1) I would like you to also mention other disorders associated with burnout, such as psychological and physical disorders, in the introduction. This would better frame the rationale for the association with hypertension. In this regard, I suggest the following references:  

doi: 10.1371/journal.pone.0185781.

  • DOI: 10.3390/healthcare10081370
  • doi: 10.1080/10253890.2018.1433161.
  •  

2) I would like you to indicate why among the various possible methods for measuring burnout you have chosen the scale you have adopted.

3)At the end of your manuscript, you rightly point out the strengths of your study. I would also like you to underline the weaknesses with a special section: "limitations", in which, for example, you talk about the size of the sample analyzed for the various types of work.

However, apart from these minor revisions, I congratulate you on your very interesting manuscript.

Kind Regards
